# In Search of Clinical Markers: Indicators of Exposure in Dampness and Mold Hypersensitivity Syndrome (DMHS)

**DOI:** 10.3390/jof9030332

**Published:** 2023-03-07

**Authors:** Kirsi Vaali, Kingsley Mokube Ekumi, Maria A. Andersson, Marika Mannerström, Tuula Heinonen

**Affiliations:** 1Department of Pathology, Medicum, University of Helsinki, 00290 Helsinki, Finland; 2Department of Food and Environmental Sciences, Biocenter 1, Viikinkaari 9, Helsinki University, 00014 Helsinki, Finland; 3Department of Civil Engineering, School of Engineering, Aalto University, 02150 Espoo, Finland; 4Faculty of Medicine and Health Technology, Tampere University, Arvo Ylpön Katu 34, 33014 Tampere, Finland

**Keywords:** mycotoxins, immunoglobulins, IgD, fecal, basophils, dampness and mold hypersensitivity, sick-building syndrome, calprotectin

## Abstract

Potential markers were sought to diagnose mold hypersensitivity. Indoor air condensed water and human macrophage THP-1 test were applied to evaluate the buildings. Basophil activation tests (BAT) were conducted and mold-specific immunoglobulins (IgE, IgG, IgA, and IgD) were measured in study subjects’ serum and feces. Exposed subjects reported markedly more symptoms from occupational air than controls. Basophils from exposed subjects died/lost activity at 225 times lower concentrations of toxic extracts from the target building than recommended in the common BAT protocol. Fecal IgG and IgD levels against *Acrostalagmus luteoalbus* and *Aspergillus versicolor* produced receiver operating curves (ROC) of 0.928 and 0.916, respectively, when plotted against the inflammation marker MRP8/14. Assaying serum immunoglobulin concentrations against the toxic *Chaetomium globosum (MTAV35)* from another building, a test control, did not differentiate study individuals. However, if liver metabolism produced the same core molecule from other *Chaetomium globosum* strains, this would explain the increased response in fecal immunoglobulins in the exposed. The altered immunoglobulin values in the samples of exposed when compared to controls revealed the route of mold exposure. The toxicity of indoor air condensed water samples, BAT and serology confirmed the severity of symptoms in the target building’s employees, supporting earlier findings of toxicity in this building.

## 1. Introduction

Few methods are in use for the clinical diagnosis of indoor mold-induced morbidity. Novel assays were developed to identify which would be suitable for the diagnosis of dampness and mold hypersensitivity syndrome (DMHS) patients. Mold exposure has often been proposed to evoke allergies. A patient’s allergic status, a pure classic type I allergy, is most easily defined in a clinical situation where there are symptoms in combination with increased amounts of IgE class antibodies and/or positive skin prick tests (SPT). However, in DMHS, the difficulty lies in the fact that specific immunoglobulins (Igs) targeting various molds are not commercially available. This means that it would be necessary to study larger groups of mold-exposed subjects versus controls. Moreover, until today, in Finland, the IgE-mediated allergy to fungal allergens has not been regarded as a common occurrence [1,2].

Basophils are located anatomically in the areas where the initial inhalational exposure occurs (airways, tonsils) and these cells become activated via IgE and non-IgE mechanisms. Therefore, basophils could play an important role in DMHS patients. Promising results have emerged from the application of the basophil activation test (BAT) in the diagnosis of drug and food allergies [3,4,5]. A form of non-IgE activation can be induced by *N*-formyl-methionine-leucine-phenylalanine (fMLP), which is a highly potent chemotactic peptide. fMLP is commonly used in studies of activated innate immune defense cells such as neutrophils or macrophages [6].

How basophils eventually become activated has not yet been elucidated and their immunological properties are still something of a mystery. In fact, there is no consensus on how human basophils should best be detected [7]. The “IgD-armed” basophils may function as circulating sentinels capable of triggering rapid innate and adaptive immune responses after sensing pathogens in the upper respiratory tract [8], but as stated, the immunological role of IgD is far from clear. Basophils and B cells are also cells rich with IgD antibodies on their surface. It is known that IgD stimulates basophils through a calcium-flux receptor that induces many cytokines, and IgD-stimulated basophils produce antimicrobial factors [8]. IgD is also a mucosal Ig, mainly present in the airways, saliva, and tear fluid. In contrast, there is very little IgD present in serum; IgD represents only 0.25% of all serum Igs [9]. IgA and IgD are both divalent Igs that can exist in secreted or membrane-bound forms. IgD is often bound to mast cells, basophils, and upper airway plasmablasts [8]. It is known that IgD reacts to the presence of colonizing respiratory pathogens [10] and the level of IgD is often elevated in patients with autoimmune diseases [10] with this Ig being released into respiratory, salivary, and lacrimal secretions. It is more unusual that serum IgD levels are elevated [11].

As basophil activation test (BAT) is the only non-IgE-mediated test method that can be used without the requirement of exposing the patient to an allergen, its use as a new diagnostic allergy test method is becoming more popular. There are commercially available pretested BAT allergen preparations with which the analyses can be performed quickly, or it can even be applied with custom-made antigenic preparations. Unfortunately, there are only four types of mycological preparations available from the sole commercial producer of antigens: *Alternaria tenuis, Aspergillus fumigatus, Cladosporium herbarum,* and *Penicillium notatum*. However, intention was to use the BAT in the assay of purified mold strains isolated from the target building, and not simply to test some commonly found mold strains.

Allergology studies do not examine inflammation markers in diagnostics, but it may be questioned, since DMHS patients show symptoms of inflammation, whether these could correlate to various clinical markers. One of them could be the heterodimeric complex of myeloid-related protein (MRP8/14, serum calprotectin), a marker of myeloid cells; its levels are increased in acute coronary artery inflammation [12,13]. Not only does MRP8/14 have antimicrobial properties, it also effectively recruits neutrophils and monocytes to the site of inflammation. In our experience, when acutely examined, the DMHS patients seem to have increased numbers of monocytes in their blood (unpublished results).

Additionally, four different mold-specific Ig classes were tested in both sera and fecal samples, as a way of evaluating the systemic and intestinal effects of molds. In publications investigating mold exposure, serum mold-specific IgG and IgE responses have mostly been in use; in fact, there are no reports describing mold-specific serum and fecal levels of IgG, IgE, and IgA, let alone IgD in mold-exposed subjects. Fecal samples were studied since the levels of Igs in fecal samples could be indicators of how mold toxicity affects the intestine and liver.

## 2. Materials and Methods

### 2.1. Study Subjects

The study population consisted of 32 subjects (12 females and 20 males) from the target, sick building and 18 subjects from the control building (13 males and 5 females). The study persons from the target building had health complaints. This led to a mapping of the diversity of the culturable indoor fungi and their production of bioactive metabolites in the target building as documented in Salo [14,15,16]. The controls were recruited from the same academic campus.

### 2.2. The Studied Buildings

The target building was a brick building, built between 1959 and 1967 and renovated in 1997–2000. It contained exceptionally high levels of toxin-producing mold strains [14,15,16,17]. Non-toxin-producing mold strains were also found in the target building, but those were not studied further here. Instead, different highly toxic strains were focused upon: *Penicillium expansum P61, Aspergillus versicolor MH33, Trichoderma viridescens Sip32, Acrostalagmus luteoalbus POB88, Chaetomium globosum MTAV3*, and *Trichoderma atroviride 14/AM*, later designated as *Trichoderma* sp. The toxicity of these microbes had been evaluated in a battery of primary eukaryotic cell cultures (Table 1). The *C. globosum MTAV3* was used as a control; it had been isolated from another building. However, another highly toxic *C. globosum* strain, *MH5*, was found to be present in the target building [17].

The control building was located 1 km away from its sick counterpart in the same academic campus. It was also a brick building but was built in 2005. It was regarded as a control building since some of the employees from the target building were able to work in the control building.

### 2.3. Sampling of the Indoor Air Condensate Water: Evaluation of Exposure Risks

The device and the technique to collect indoor air water samples with the so-called E-collector have been described elsewhere [22,23,24]. Briefly, the principle of the collection is based on the phenomenon of the condensation and subsequent freezing of water molecules on the top of cold surfaces of metal plates. After melting of the frozen water, the condensate was collected from the tray of the device into glass tubes that were transferred to the laboratory where they were subjected to the cytotoxicity assay using human macrophages (derived from THP-1 monocytes). This collection technique enables harvesting of airborne toxic substances in indoor air vapor that may contain toxins [15] as well as various large molecules [25]. This is a newly globally patented technique: (US Patent 10,502,722 B2; CA patent 2,972,162; FI 128773 B; EU 3241010). For estimates of exposure levels, the relative humidity (RH%) and temperature (°C) were recorded from the building and the amount of water inhaled over the course of time was calculated [26]. However, it must be noted that humidity and, consequently, water-borne toxicity often vary seasonally.

### 2.4. Cytotoxicity of Indoor air to THP-1 Macrophages

The toxicity of the indoor air water condensates collected from different locations of the buildings was evaluated with the human monocytic leukemia cell line (THP-1) using the water-soluble tetrazolium salts (WST-1) assay [24,27]. The WST-1 assay is an indicator of mitochondrial activity and a commonly used technique to assess cell viability; a decline in mitochondrial activity reflects a loss of cell viability, i.e., cell death. In contrast, elevated mitochondrial activity is indicative of an increased cell number, i.e., proliferation, or increased cellular respiration induced by mitochondrial uncoupling reactions [28,29]. The WST-1 Cell Proliferation Reagent was obtained from Roche (Basel, Switzerland). Human THP-1 monocytes (Cat. No. TIB-202) were from ATCC (LGC Promochem AB, Boras, Sweden), and were verified to be *Mycoplasma*-free (MycoAlert™ kit, Lonza Basel, Switzerland) prior to use. THP-1 monocytes were maintained in RPMI 1640 Medium supplemented with 10% fetal bovine serum (FBS), and differentiated to macrophages, with 25 nM phorbol 12-myristate 13-acetate (PMA) (Sigma Aldrich, Steinheim, Germany) for 48 h followed by a 24 h recovery period.

The cells were seeded into 96-well plates at a density of 10^5^ cells/well and exposed for 24 h to the indoor air condensates at 10% concentrations (in RPMI supplemented with 5%FBS). Cells exposed to an equal volume of distilled water (10%) served as negative controls, and nickel II sulfate hexahydrate (2.0 and 20.0 µg/mL) was used as a positive control. All samples and controls were tested in 6 replicates. In the assessment of cell viability, WST-1 reagent was added at 10 µL/well to the cells for 3 h, and subsequently absorbance was read at 450 nm. The absorbance is directly proportional to the mitochondrial activity (cell viability). The absorbances were normalized, i.e., the untreated control was set as 100%, and all other data were calculated relative to the control absorbance as either % decrease in cell viability (negative values) or % increase in proliferation (positive values). The statistical significance of the changes as compared to the untreated control was tested using Student’s *t*-test.

### 2.5. Extraction and Determination of Protein Content of Purely Cultured Mold Strains

The purely cultured and identified mold strains were extracted into solution buffer. The samples were processed by scraping the pure culture from the plate into homogenization tubes with protease-containing extraction buffer (Pierce EDTA-free protease cocktail-1% DMSO-10 mM PBS pH 7.4).

Silica sand 0.2 g/2 mL tube (BioSpec Products, Bartlesville, OK, USA) was added to the mold preparations. The diameter of the sand was 0.1 mm. Samples were homogenized by shaking with FastPrep-24 MP homogenizer (MP Biomedicals, Auckland, New Zealand) for 5 × 20 s every 5 min at 5.5 power. The samples were then centrifuged, and the supernatant was assayed for their protein content (BCA, Pierce, Waltham, MA, USA) using the microtiter plate method. In the BAT test, the extracts were used at various concentrations (tested from 0.1 ng/mL to 1 µg/mL). In the immunoglobulin (Ig) assays, ELISA plates were primed with 2 µg/mL protein of the studied mold extract.

### 2.6. The Basophil Activation Test (BAT) Method

In the flow cytometric analysis of basophils, the detection technique assesses not only the shape and size of the cells, but also the fluorescent antibodies used for labeling on the surface of the cells. In the isolation of the basophils and for the BAT, a ready-made kit provided for clinical laboratories was used (produced by Bühlmann Laboratories, Basel, Switzerland). The kit contains prelabeled CCR3 and CD63 antibodies, IgE and non-IgE-mediated controls (FcεRI and fMLP), and reagents and instructions for the entire work protocol. Two different fluorescent labels were used to stain cellular structures. The first fluorescent dye binds to the CCR3 molecule on the cell’s surface; it targets the basophil population, and this is PE-labeled. The second fluorescent dye detects the so-called CD63 structure (labelled with FITC), a structure that increases on the surface of the basophil when the cells become activated [30]. In this case, the antigens crosslink the immunoglobulins on their receptors, activating the cell and this models the in vivo reaction. Both IgE cross-linking FcεRI (IgE-mediated control) and fMLP (*N*-formyl-methionyl-leucyl-phenylalanine, non-IgE-mediated control) were in use. fMLP mimics the effects of microbial stimuli and both positive controls were used in the implementation of the test in each subject. Subjects have different IgE-mediated activation capacities; for example, there are individuals who have a too low activation percentage (<5%) for the IgE-mediated control sample. Such subjects are not appropriate for evaluation with the IgE-mediated part of the assay, (approximately 8% of the general population), but for them, fMLP can still be applied as a positive control.

### 2.7. Full Blood Sample Processing for BAT Assays

Potassium-EDTA venipuncture blood samples were drawn (3 mL). The BAT tests were performed on the blood samples in a laminar hood to prevent any unwanted contamination and according to the instructions provided by the producer (Bühlmann Laboratories). Before the actual study was performed, the BAT assays were pretested with four commercial mold preparations (*Penicillium notatum, Cladosporium herbarum, Aspergillus fumigatus and Alternaria tenuis* from Bühlmann Laboratories, Basel, Switzerland) and with basophils from four subjects who were known to have experienced symptoms, and who had been exposed to various strains of molds. The commercial mold preparations were pre-titrated to the correct concentration by the producer, i.e., 22.5 ng of the mold protein preparation/test tube stimulation. Only freshly reconstituted commercial mold preparations were in use. The target building mold extracts were assayed in different dilutions: 0.1, 1, 10, 100 and 1000 ng of protein/test tube.

Data acquisition and analysis were carried out in a CyAn ADP flow cytometer (Beckman Coulter, CA, USA). Appropriate compensations were undertaken according to Bühlmann Laboratories BAT kit’s instructions. Activation of basophils was assayed in each subject’s control, FcεRI, fMLP, and mold preparation tubes as the percentage of the activated, CD63 gated positive cells in the upper right quadrant.

### 2.8. Collection of the Full Blood, Sera, and Fecal Samples

Full blood was collected from the study persons in 5 × 9 mL serum tubes (without additives) and samples were transported to the laboratory within 2–3 h of sampling. They were centrifuged at 3000× *g* for 15 min, divided into portions and frozen at −80 °C. Fresh full blood was used for BAT testing directly after its arrival in the laboratory.

During the following weeks, study subjects transferred their fecal samples directly after sampling to −20 °C storage. They were provided a sample tube (Sarstedt, Numbrecht, Germany 80.623.022) for the transport of the sample and a polystyrene foam pack from Sarstedt (Numbrecht, Germany), which were frozen before transportation to the study building (Sarstedt, Numbrecht, Germany 95.1123, polystyrene foam box and cold transport container). Samples were then collected from the building’s freezer and transported to the laboratory in dry ice. In the laboratory, the fecal samples were stored at −70 °C until assay.

### 2.9. Homogenization of Fecal Samples for Ig Assays

Samples were weighed and a precalculated amount of extraction buffer (1.00 g of fecal/3 mL of buffer) was added. The buffer contained Pierce’s cocktail of EDTA-free protease inhibitors dissolved in 1% DMSO—10 mM PBS, pH 7.4. Thus, the fecal content was 1.66 µg/µL of buffer. The samples were homogenized on a FastPrep^®^-24 (Fort Lauderdale, FL, USA) using silica sand, 0.1 mm in diameter with 0.2 g of silica sand in each sample (BioSpec Products, Bartlesville, OK, USA). Homogenization was performed for 5 × 20 s for 5 min at 5.5 power and samples were cooled with ice between the homogenizations. The samples were then centrifuged, and the supernatant was used for the assay of microbial-specific immunoglobulins (IgG, IgE, IgA, and IgD) at several different dilutions, starting from a maximum concentration of 100 µg/mL, e.g., fecal contents were 100, 33, 10, and 0.3 µg/mL. Later, some tests were performed with 333 and 1000 µg/mL in studies to examine the receiver operating characteristic curve ROC.

### 2.10. Microbial-Specific Ig Assays in Serum and Feces

The assay was performed by the coating of the mold extracts with a 2 μg/mL protein preparation in 50 µL of PBS at pH 7.4 overnight at +4 °C. To prevent nonspecific binding, titer plates were blocked with 3% BSA in 150 µL PBS at room temperature for 2 h. The titer plate wells were then washed three times with 300 µL of 10 mM–0.05% Tween 20 in pH 7.4 phosphate buffer. Antibody assays were performed at 2–4 different dilutions depending on which Ig was being assayed. The incubation time of the samples (serum or fecal extracts) was overnight at +4°C to obtain a uniform result. The antihuman HRPO conjugates for IgG, IgE, IgD, and IgA immunoglobulins (Southern Biotech, Birmingham, AL, USA) were diluted 1:4000, 1:1000, 1:4000, and 1:4000, respectively, and they were assayed after an overnight incubation (at +4 °C). Before the color reaction, the titer plates were washed five times with phosphate buffer and 100µL TMB was added for detection of the substrate (Ultra TMB-ELISA, ThermoScientific, Waltham, MA, USA), for 15 min in IgG plates and for 30 min for all other subclasses of Ig. The TMB reaction was quenched with 100 µL of 1 M HCl, with the results being measured at 450 nm (Hidex Sense microtiter reader, Turku, Finland).

### 2.11. Sensitive C-Reactive Protein (CRP), MRP8/14, and Fibroblast Growth Factor-21 (FGF-21) in Serum Samples

The CRP assays were purchased from the commercial clinical Vita Laboratory (Helsinki, Finland). Myeloid-related protein (MRP8/14) was quantified with the commercial clinically validated ELISA kit of Bühlmann Laboratories AG, (4124 Schönenbuch, Switzerland). Human FGF-21 was assayed by the commercial ELISA kit of BioVendor (621 00 Brno, Czech Republic). Both assays were measured in a Hidex Sense microtiter reader (Turku, Finland).

### 2.12. MRP8/14 (Calprotectin) Assays in Fecal Samples

The fecal tubes were thawed at room temperature and samples were taken with a CalexCap (Bühlmann Laboratories, Basel, Switzerland) fecal sampling device for the extraction of calprotectin from the specimen. The samples were incubated at room temperature overnight according to the instructions (EK-CAL2-WEX, Bühlmann Laboratories, Schönenbuch, Switzerland).

### 2.13. Receiver Operating Curve (ROC)

The ROC graph is used in clinical laboratory analysis to indicate how well a marker performs, e.g., the extent to which it reflects a clinical symptom. The closer the calculated value reaches an absolute value of 1, the better is the predictive value of this marker. The two assay components are (1) assay specificity (accuracy) and (2) sensitivity. When the diagnostic test is a continuous variable, the desired cutoff value can be applied to it, which then classifies the subjects as either healthy or sick. ROC Graphs were analyzed with the GraphPad Prism program (San Diego, CA 92108, US).

### 2.14. Statistical Methods

From the assay results, differences between the two groups of subjects’ immunoglobulin scores were calculated using the Student T test and the nonparametric Mann–Whitney test with the GraphPad Prism program. At each dilution, only the exposed vs. controls were compared.

## 3. Results

### 3.1. Symptoms Reported by the Study Subjects

Unfortunately, some subjects were so severely ill that they could not be included into this study as they had left their place of employment. Some of the exposed subjects had been working in the target building for some decades. However, it is possible that the toxic indoor air had developed in rather recent times. One problem in this kind of an occupational indoor air study is that the control or the exposed subjects might be living in a domestic residence contaminated by mycotoxins. See results in Table 2.

### 3.2. Indoor Air Condensate Water Samples Show THP-1 Cell Toxicity in the Target Building but Not in the Control Building

The toxicity of the indoor air condensate water samples as evaluated using in vitro THP-1 tests was classified with the following empirical criteria: based on over 3000 assays from indoor air water samples, it can be suggested that if the results are expressed as ED_50_, then these are too high to allow the reliable estimation of human exposure in THP-1 tests, and instead the ED_10_ value should be applied [24]. We suggest that when the number of THP-1 cell line in 24 h cultivation with the 10 µL water sample in the total volume of 100 µL in a microtiter plate undergoes a cell death of 0–3%, this does not indicate the presence of toxic conditions in a building, irrespective or not whether the result reaches statistical significance. When there is 3–5% cell death, and it is statistically significant, the indoor air is somewhat toxic, i.e., indicative of the presence of slightly harmful toxicity in the building. Based on the previously published reports [22,23,31] and according to our unpublished clinical experience, we estimate that if the indoor air water samples evoke a 5–10% and statistically significant cell death at a 10% sample concentration, then this is a clear signal of the presence of toxic indoor air. Thus, it is our belief that when there is (statistically significant) >10% cell death, the indoor air should be regarded as highly toxic, and the occupants should vacate the building immediately. In this study, two different building inspectors sampled the indoor air water in the toxic building (Table 3), whereas one inspector sampled the control building (Table 4).

### 3.3. In the BAT Test, a Negative Result was Obtained from Almost All Mold-Exposed Subjects

Only 2/31 subjects from the target building cases provided an activation response in the BAT for the six tested mold strains and for their three combinations (Figure 1). In particular, the *Penicillium expansum* responses were too low to be measured, suggesting that cells either died or were inert due to tested strain’s toxicity. A retesting with different dilutions was performed with 0.1, 1, 10, 100, and 1000 ng of mold protein/test tube, but even the lowest concentration, being 225 times less than that recommended by the producer, did not produce activation. Due to the negative results, the control subjects were not studied with the BAT.

### 3.4. Studies of the Ig Levels in Serum and Fecal Samples

Therefore, studies were proceeded to evaluate the systemic and intestinal immunological status of the study persons using these six different mold strains by examining serum and fecal antibodies (immunoglobulins, Igs, Figure 2, Figure 3, Figure 4 and Figure 5). Determination was performed to evaluate if the exposed differed in their Ig responses in comparison to controls.

In serum, all mold strains showed an increased IgE response in the exposed vs. controls except for *C. globosum MTAV35*; its levels did not differ between the two groups (Figure 4 and Figure 5).

### 3.5. Correlation of the Serum vs. Fecal with Respect to Either an Increase or Decrease of the Ig Responses

This Table 5 is constructed to help the reader to visualize the results of all the Ig findings. The basic concept is to reveal whether the exposed subjects had a greater or reduced Ig response than the controls in serum or fecal samples.

### 3.6. Inflammatory Markers: CRP, serum MRP8/14, and Fecal Calprotectin

The liver-derived acute-phase protein of hepatic origin, C-reactive protein, was evaluated (CRP has been by far the most common inflammatory marker in clinical use); levels higher than 0.1 mg/L could be determined in the assay. However, MRP8/14, also known as serum calprotectin, is a more sensitive marker since it is not synthesized by the liver, and thus it is a more rapidly responding marker of acute inflammation. The monomers of MRP are prestored inside cells of myeloid origin, allowing the cells to directly release them, whereas CRP is produced by a complicated synthetic process in the liver, which requires a time as long as 15 h.

There was no difference in the serum CRP, MRP8/14, or fecal calprotectin levels in the exposed subjects as compared to the controls. However, interestingly, mold-exposed men showed significantly reduced MRP8/14 levels (in the figure marked as serum calprotectin results in Mann–Whitney nonparametric test, *p* = 0.0145, Figure 6).

### 3.7. Mitochondrial Dysfunction Marker FGF-21

As fatigue is expected to be one of the main symptoms in severe mold-exposure, it was anticipated that there could well be a difference in the levels of the mitochondrial marker FGF-21. However, there was no significant difference between the mold-exposed study subjects and controls.

### 3.8. Receiver Operating Curve (ROC) and the Ratio of Serum MRP8/14 (Calprotectin) to Fecal Immunoglobulins

The most significant results of the fecal Ig levels with the serum MRP8/14 values were compared from the same subjects. Serum MRP8/14 was chosen as the inflammatory marker for comparison, as it clearly delivered a high score above the cutoff value (0.5 µg/mL) and was more sensitive than CRP or fecal calprotectin. In the comparisons presented below (Figure 7), the Ig results that differed most in the results between the study groups were selected.

Since fecal extracts have not been studied with ELISA methods in general, whether a higher concentration of the microbial extract would improve these fecal ROC results was tested in the coating (Figure 8). When the protein extract of *A. versicolor* was coated with 100 µg/mL and the IgD responses of the fecal extract were tested, the ROC curve was 0.723. The respective results from fecal *A. versicolor*-specific IgD and serum MRP8/14 provided a ROC value of 0.916 when the mold extract was increased to 1000 µg/mL and 0.863 when the coating level was 333 µg/mL. However, as the separation was based on the microbe-specific Ig results, and MRP8/14 levels played a rather modest role in general, it does seem probable that these results would not be helpful in the diagnosis of mold exposure in the future.

## 4. Discussion

This study aimed to develop assays for the clinical diagnostics of dampness and mold hypersensitivity syndrome (DMHS). The approach was to examine isolated and pure cultured toxic molds of a “sick building”, the immunological response of the mold-exposed vs. controls, and furthermore not simply to use some generally well-known mold preparations. Based on earlier publications, the presence of molds is not the issue; instead, it should be the existence of fungal metabolites excreted by actively growing molds, possibly including mycotoxins, biosurfactants, and enzymes [14]. According to the results published in a doctoral dissertation made from the same target building, the vast majority, i.e., >70% of the tested microbial colonies were toxic when examined in cell tests [14].

According to the results presented here, the studied building was clearly toxic in many respects: severe symptoms reported by >90% of the employees, which were confirmed objectively in various in vitro tests.

### 4.1. BAT Results Confirm the Toxicity of the Studied Mold Strains

The promising results emerging from food allergy and non-IgE-dependent basophil activation tests (BAT) formed the basis for decision to utilize the BAT in this hotly debated topic of how best to assess poor indoor air quality. Furthermore, in the earlier study of Mirkovic et al., BAT discriminated *A. fumigatus*-sensitized from non-sensitized cystic fibrosis patients. The basophil activation marker used here (CD203c) was different from that used by Mirkovic (CD63), this technical difference is not responsible for the major differences between our results and those of Mirkovic et al. [32]. Both markers can be equally applied according to the manufacturer of the test kit in the evaluation of basophil activation when analyzed using flow cytometry.

It was only later that susceptions rose of the extremely toxic nature of the strains that were examined, and that basophils cannot become activated if they are dead, although this is not evident in the flow cytometric analysis; cells are assessed similarly in flow cytometric assays irrespective if they are dead or alive. In contrast, basophils can be reliably detected based on their constitutive eotaxin receptor, CCR3, but antigen stimuli induces an activation signal and the appearance of the CD63 on the cell membrane surface of the basophils. This phenomenon is not expected to occur in dead cells. The preliminary tests with commercial preparations (mold preparations produced by Bühlmann Laboratories) were all successful, probably as these preparations were not isolated from toxic strains. The suspicion was that the very high toxicity of mold preparations from the target building would be the reason for the negative results in BAT. Nonetheless, it does seem plausible that BAT still has a practical function: it could be a quick test when there is need to know if the isolated strains are toxic for humans. The benefit is that in the BAT test, the subjects being evaluated need not be exposed to these toxic preparations as a part of the test.

### 4.2. Evidence That the Study Subjects in the Target Building Are Being Exposed more to Bioactive Fungal Metabolites Than Those Working in the Control Building: Indoor Air Water Kills Human Macrophage Cells

The earlier research results that show hazardous exposure from the target building were based on the test results and secretions from pure cultured molds, and these results were obtained with sperm and kidney cell cultures (Table 1). In these earlier methodological approaches, the pure cultured microbial samples originated from the building, and the test cells were not obtained from the exposed study subjects but from animals (porcine sperm or kidney epithelial cells, or murine neuroblastoma cells, Table 1). The collected water samples condensed from the indoor air were tested with a human macrophage cell line (THP-1), in order to examine how the human immune system reacts to the toxic molds and this was measured in response by means of the WST-1 assay, which assesses mitochondrial toxicity (Table 3 and Table 4) [24]. There is a recent publication describing WST-1 results of 688 indoor air water samples [24] in addition to unpublished results from Spring 2021 at FICAM (Tampere University). From these results, we have drawn rough estimates that a 3–5% THP-1 cell death is indicative of somewhat harmful indoor air, 5–10% death markedly harmful indoor air, but if >10% THP-1 cells die, then the building’s residents should be evacuated as soon as possible. However, the relative humidity (RH%) and indoor air temperature influence the severity of the toxicity; if the room is very moist and the temperature is high, then the effects of exposure are markedly intensified. As far as we are aware, this is the only assay that takes several different toxic factors into account in the evaluation of indoor air quality [31]. The results from this study are in line with earlier findings showing that the ambient air present in a mold-contaminated building contains cytotoxic substances (Table 1, Table 3 and Table 4). Indeed, by measuring how THP-1 cells respond to the water condensate from the indoor air is a good way to screen the air quality and identify those buildings that are markedly toxic. The first publication of this method revealed that the majority of the buildings tested did not require any renovation actions [24].

### 4.3. Results of Mold-Specific Immunoglobulin (Ig) Assays in Serum and Fecal Samples

Various Ig subtypes were studied in serum and fecal samples to see if they could separate exposed subjects from controls. At a first glance, the sera vs. fecal results do not provide any clear logical outcomes with respect to these six different mycotoxin strains. These results could be interpreted in a roundabout way: the results suggest that Igs may be markedly reduced in the exposed compared to the controls, when especially depleted from those tissues, which are not their optimal territory. For instance, the fecal levels of IgE in the exposed subjects are lower than in controls (Figure 3 and Figure 5), and since IgE is not primarily involved in the defense system of the intestinal tract, the nonsignificant results could be expected. The results of IgA determinations are according to the general understanding: gut is the primary area for IgA production and therefore these mucosal immunoglobulin amounts should be increased in fecal samples of the exposed subjects as compared to the individuals working in healthy buildings. The reduced levels of IgA in the fecal samples of the exposed subjects could be explained by the first pass metabolites, which are now different from the original structures.

In subjects exposed to *P. expansum,* the amounts of IgA and IgG were decreased in serum (Figure 2) and increased in fecal samples (Figure 3) in comparison to controls. It is intriguing that there were no differences between the two study groups in the serum levels of IgG against the highly toxic *P. expansum*. It could be speculated that this strain is the most harmful for the body after its excretion into the intestinal tract in the liver phase I and phase II metabolism. After metabolism in the liver, the toxins produced by *P. expansum* are possibly converted into a more toxic form and need to be totally and safely excreted from the body, and therefore, this will be reflected as an increase in fecal IgG and IgA levels of the exposed subjects.

*A. versicolor* grows in wallboards, insulation, textiles, ceiling tiles, and manufactured wood and is often found in water-damaged building materials; furthermore, it may flourish on nutrient-poor materials [33]. *A. versicolor* is known to gain access to the human body by passing through airway epithelial cells and it can evoke invasive alveolar aspergillosis [34,35]. When the fecal samples show a decrease in *A. versicolor* Igs in the challenged subjects, this could mean that detoxification during the first pass through the liver could have lessened the impact of its toxicity. Since both serum and fecal IgD levels against *A. versicolor* were reduced in exposed subjects (Figure 2 and Figure 3, respectively), it could be speculated that the body requires the immunological protection conferred by this immunoglobulin in its barrier sites such as airways, nasal passages, or ocular locations, and IgD moves away from the vascular and gastric systems. The most significant decrease in exposed subjects was observed in the levels of *A. versicolor*-specific fecal IgD (*p* < 0.0001).

*Acrostalagmus luteoalbus*, which is a deep sea-derived fungus found recently in Korea and reported to be rare [36,37], was a novel finding in this target building [18,38]. These recent publications investigated the same building as examined in our study. Andersson et al. reported how the major constituent of the building’s microbiota was *A. luteoalbus;* this species was found in settled dust in rooms and in the damp cork lining of the outdoor wall. Its mycotoxins have been identified as melinacidins II, III, and IV [18]. Melinacidins produce epipolythiodioxopiperazines (ETPs), i.e., their toxic metabolites are produced by methylation and sulfation reactions; these enzymes are needed for methylations of genes and are critical for human health. Saffron plants may be contaminated by *A. luteoalbus* [39], but saffron is hardly ever used in Finnish cooking or in purchased food items. *A. luteoalbus* was somewhat special in that it produced a clear increase in the fecal IgG level of exposed subjects, suggesting that its hepatic first pass metabolites were responsible for this finding. It can be suggested that the negative air pressure due to the building’s ventilation had sucked the mycotoxins out from the cork board and stimulated the exposed subjects to produce immunoglobulins against *A. luteoalbus* [38].

Serum IgE and IgA levels for *A. luteoalbus* were significantly increased in exposed subjects when compared to the controls (*p* < 0.05 and *p* < 0.001, respectively) (Figure 4). However, most interestingly, the fecal *A. luteoalbus*-specific IgG values were clearly elevated in exposed subjects (*p* < 0.0001), but there was no significant difference in serum IgG levels in comparison to the respective control group (Figure 4). Furthermore, *A. luteoalbus*-specific IgA levels were increased at the lowest fecal concentrations (*p* < 0.0001) (Figure 5). This could be interpreted as both serum and liver forms of *A. luteoalbus* being potentially harmful for the body but that the liver metabolism produces a potent toxin from *A. luteoalbus*.

Although none the serum levels of *C. globosum*-Igs differed from those measured in the controls (Figure 4), the fecal IgG levels were increased (Figure 5). Importantly, it should be noted that even though the highly toxic, reference strain of *C. globosum MTAV35* that was studied previously [20,21], was not the same strain that was present in the target building, it does provide some valuable information: (1) One could argue that the Ig results do provide a correct perspective, and that the serum Igs do not recognize *MTAV35* strain, as the exposed subjects did not differ from controls. It was known that there were other *C. globosum* strains in the target building that were not studied. (2) However, after the body has been exposed to the other strains of *C. globosum*, then the liver-derived metabolites of this microorganism can trigger the same kind of Ig response as that produced by the building-derived *C. globosum* strains. Therefore, *C. globosum* derivatives could be extremely toxic to the liver and induce the increase seen in fecal IgG levels as observed in the exposed subjects. Nonetheless, it is difficult to explain why there were no similar alterations in the fecal IgA concentrations.

In serum, the concentration of *Trichoderma* sp. *(=atroviride 14/AM)* specific IgE was significantly increased in the serum of the exposed subjects as compared to controls (Figure 4). Instead, the levels of fecal *Trichoderma* sp. *(=atroviride 14/AM)* Igs were decreased in nearly all the studied Ig subclasses in the exposed subjects when compared to controls (Figure 5). This can be interpreted that for the defense of body, there is less of a need for intestinal territory protection. The results of *Trichoderma viridescens* are difficult to explain when the changes in the IgG levels in both serum and fecal specimens were nonsignificant whereas the fecal IgD levels were increased in the exposed subjects (Figure 2 vs. Figure 3).

IgD is found on the surface of the majority of B lymphocytes [40]. Since IgD receptors seem to disappear from B cells when they undergo maturation prior to secretion [40], this would explain why mainly a decrease in IgD values were detected simultaneously with increases in those of IgG or IgE. In wild-type mice, a class-switch recombination for IgD production appeared in nasal-associated lymphoid tissue and was microbiota-dependent [41]. In humans, IgD positive plasma cells are prominent in sinus tissues and their amounts were reported to increase in patients with chronic rhinosinusitis [42]. In the exposed subjects, the decrease, or no difference in all IgD results may suggest that there had been no need for IgD defense in body locations other than in saliva, airways, or nasal secretions, but unfortunately, these kinds of samples were not collected. In particular, *A. versicolor*-specific IgD levels were significantly decreased in both serum and fecal samples from the exposed subjects.

### 4.4. Other Tested Markers for the Clinical Diagnostics

Mitochondria produce about 90% of the energy generated by cells. Mycotoxins from mold-damaged buildings have been shown to be highly potent in interfering with mitochondrial functions. The body recognizes elevations in the levels of NAD/NADH and AMP/ATP, which indicate a lack of cellular energy, initiating processes of mitochondrial biogenesis, fat burning and the production of oxygenated ATP. FGF-21 induces the transfer of fatty acids from fat cells to the liver [43]. Mitochondrial function and FGF-21 are of interest in the diagnosis of moisture-exposed subjects, e.g., the determination of FGF-21 would represent an easily measurable serum marker in individual patients. For example, the *Fusarium* mycotoxin, enniatin, has been shown to inhibit the formation of the mitochondrial membrane potential, the mechanism underlying energy production in cells [44]. Moisture damage buildings also contain bacterial toxins, e.g., amylosin isolated from *Bacillus amyloliquefaciens*. Amylosin has been shown to be a potent mitochondrial toxin capable of causing potassium ion efflux, and activation of inflammasomes [45]. Therefore, it could be expected that the levels of the mitochondrial marker FGF-21 would have been able to separate controls from cases.

Many of the published mycotoxin works have been conducted in cell culture or production animals [46,47]. These studies do not take into account the genetic profiles of exposed subjects, nor do they consider the various routes of environmental exposure. For instance, enniatin B cell arrests cell cycle at G_0_/G_1_, induces apoptosis and activates the so-called master switch of inflammation, and as with trichothecene mycotoxins, activates inflammasomes (such as the NLRP3 inflammasome) [48]. This activation can be measured as IL-1β production [49,50], which activates the cellular signaling pathways known to participate in inflammasome-induced cellular pathology: potassium efflux, augmented ROS production, and lysosomal damage. Therefore, it was tested whether a combination of an inflammation marker could be beneficial in improving the Ig results in clinical diagnostics. Indeed, when a measure of inflammation was the other factor, then the results generally exceeded the accepted ROC values (>0.85).

MRP8/14 (serum calprotectin) is an activator of Toll-like receptor 4 (TLR4) [51] triggering an increase in the numbers of neutrophils and monocytes and activation of inflammasomes (activation of the innate immunity). It is well known that males have a higher risk of suffering from cardiovascular disease than females. MRP8/14 values differed significantly in exposed and control males but not in women. MRP8/14 is not yet a commonly exploited marker in clinical cardiology, although it has been considered to be a relevant inflammation factor in the development of diabetes progressing towards cardiovascular diseases [13,52] and in juvenile arthritis [53]. In patients with SARS-COVID-19 infection receiving treatment in an intensive care unit, the ROC assay results indicated that MRP8/14 was superior to other markers (ferritin, CRP, lactate dehydrogenase) in evaluations of the patient’s health status [54]. Wirtz reported that mortality of SARS-COVID-19 patients in intensive care units was related to a reduction in MRP8/14 levels, whereas the surviving subjects had increased MRP8/14 levels [55]. This could explain the finding of lower MRP8/14 concentrations in exposed men vs. the men in the control group. Furthermore, as demonstrated in an *A. fumigatus* mouse infection model, inflammasome activation via NLRP3 eventually provides immune protection and IL-1β-mediated survival [56].

### 4.5. Strength and Weakness of this Study

The strengths of this study are that (1) four different Igs, even the rarer IgD, were studied; and (2) assays were performed in both serum and fecal samples of the study subjects. (3) Interestingly, basophil cells from exposed subjects had become inactivated by the microbial preparations isolated from the toxic building. (4) Even six toxic building-derived, purely cultivated mold strains, and not just ordinary cultivated strains, were studied. (5) The toxicity of the target building mold strains had previously been tested in various animal cells; these new results provide confirmation of toxicity when tested in primary human basophils of the exposed subjects. (6) Furthermore, the results of indoor air water, when administered to human macrophage cell line (THP-1) in vitro showed that the conditions in target building were toxic to the inhabitants, whereas the rooms examined in the large control building were not toxic. (7) High ROC scores could be obtained when the fecal IgD results of *A. versicolor* or fecal IgG of *A. luteoalbus* were combined with a marker of inflammation, i.e., serum calprotectin (MRP8/14). (8) These results suggest that IgD assays against some of the strains should be studied further in sputum, ocular, or nasal secretions. (9) This study was performed well before SARS-COVID-19 had spread to Finland, so the results are not affected by the COVID-19 pandemic.

The weakness in this study is that nasal, ocular, or sputum samples were not examined of the tested study subjects, which could have provided a deeper perspective of the IgD results. In addition, the viability of the basophils should had been confirmed after their exposure to the mold extracts.

## 5. Conclusions

If the building-derived mold preparations are highly toxic, the BAT will most probably reveal no basophil activation in any of the study subjects if the basophils die. Thus, one must find additional ways to test whether the ambient conditions in the building from which the microbial extract has been taken are toxic to the people working/living in that location. The benefit of BAT would be that in such a case, a few subjects could be tested to confirm the toxicity of the building-derived samples. It is important that the test should be conducted utilizing pure cultured mold preparations from the target buildings, and not to use common microbial cultures. Evidently, novel markers are needed for the confirmation of clinical toxicity; in this respect, it may be beneficial to perform ROC analysis utilizing a combination of approaches, e.g., assaying the levels of mycotoxin-specific immunoglobulins (Igs) combined with those of an inflammation marker such as MRP8/14. There was an increased response in exposed vs. controls in the mold-specific IgE values against all those mold strains that were from the target, toxic building, but there was no difference in serum IgE response against the *C. globosum* strain from another building. This suggests that from a methodological viewpoint, the Ig results are reliable. Interestingly, fecal Ig values indicate that when the mycotoxins synthesized by this distinct C. *globosum* strain are processed by hepatic metabolism, the metabolites formed could be as toxic as those synthesized by the strain detected in the problem building. However, the serum and fecal Ig assays can only be applied in clinical diagnostics if a sufficiently large number of exposed and controls from a few buildings can be tested, which is unpractical. More research would be needed to study the routes of exposure in indoor air toxicology, perhaps in some kind of surrogate or roundabout way, but the important exposure routes (blood, fecal, airway lavage, lacrimal, sputum samples, etc.) need to be evaluated. This roundabout way means one could reveal that the Igs in the target tissue have been relatively more depleted in exposed vs. controls, especially in those tissues where they would not be considered as the optimal Igs. The indoor air condensed water test with THP-1 cells is a relatively inexpensive method for the evaluation of hazardous indoor air quality. It can be suggested that the first steps in indoor air studies would be to evaluate the symptoms of the subjects in the suspected building and then to conduct indoor air condensed water toxicity tests. These can be followed by BAT tests in some subjects, which must be performed utilizing very large dilutions of the culture purified mold strains from the target building.

## Figures and Tables

**Figure 1 jof-09-00332-f001:**
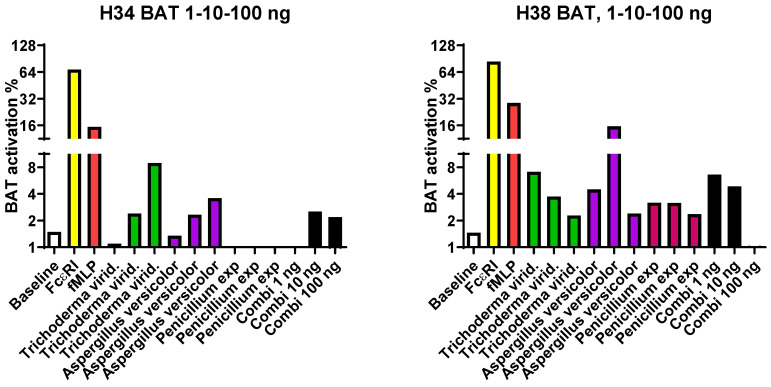
Result of two subjects’ positive basophil activation BAT tests: Baseline is without an activation stimulant. FcεRI is the IgE response positive control; fMLP is the non-IgE-mediated positive control. The protein concentrations of the stimulating microbes used here were 1, 10, or 100 ng, respectively, from each strain tested. Combi signifies the combination of all three different strains; Combi 1 ng refers to each mold strain at a final concentration of 1 ng/tube, Combi 10 represents 10 ng/tube, and in Combi 100, the final concentration of all three strains was 100 ng/tube. A positive response can be considered when the BAT activation is about 2–3 times higher than that measured in the baseline tube. One exposed subject (H34) displayed an impressive dose response to *Trichoderma*, while for the other exposed subject (H38) the same doses became too toxic. This figure presents only those two exposed subjects in whom the stimulation reaction to a mold was positive. In all the other exposed subjects, results with the six mold strains examined in this study showed no evidence of basophil activation.

**Figure 2 jof-09-00332-f002:**
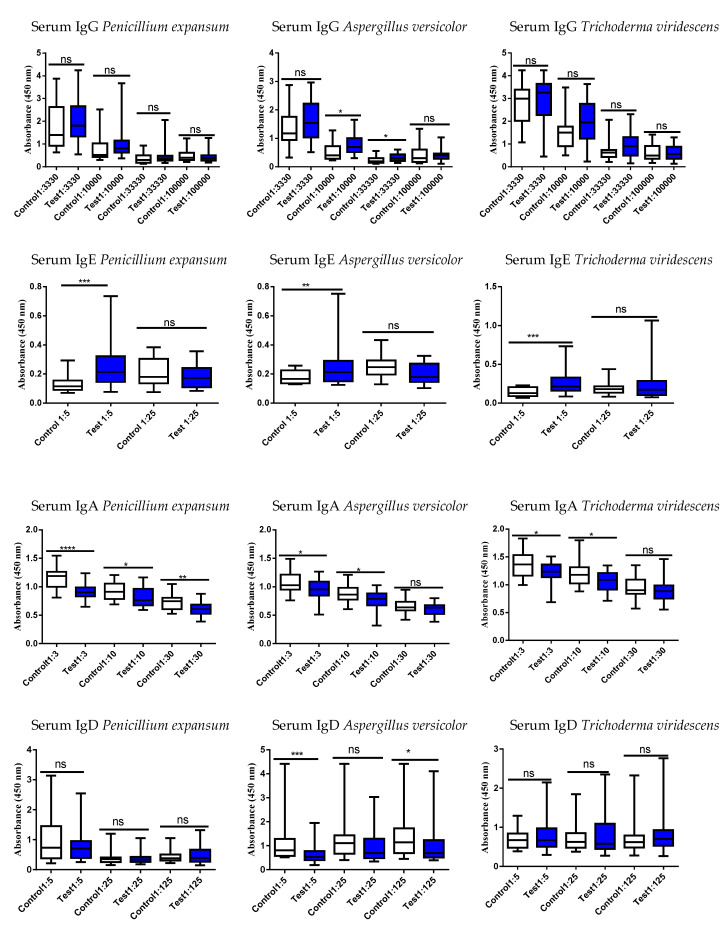
Serum immunoglobulin Ig responses against extracts of *Penicillium expansum*, *Aspergillus versicolor*, and *Trichoderma viridescens*. Boxplots indicate median values with 95% confidence intervals; statistically significant * *p* < 0.05, ** *p* < 0.01, *** *p* < 0.001 and **** *p* < 0.0001 when the study persons were compared to controls. Ns = non-significant.

**Figure 3 jof-09-00332-f003:**
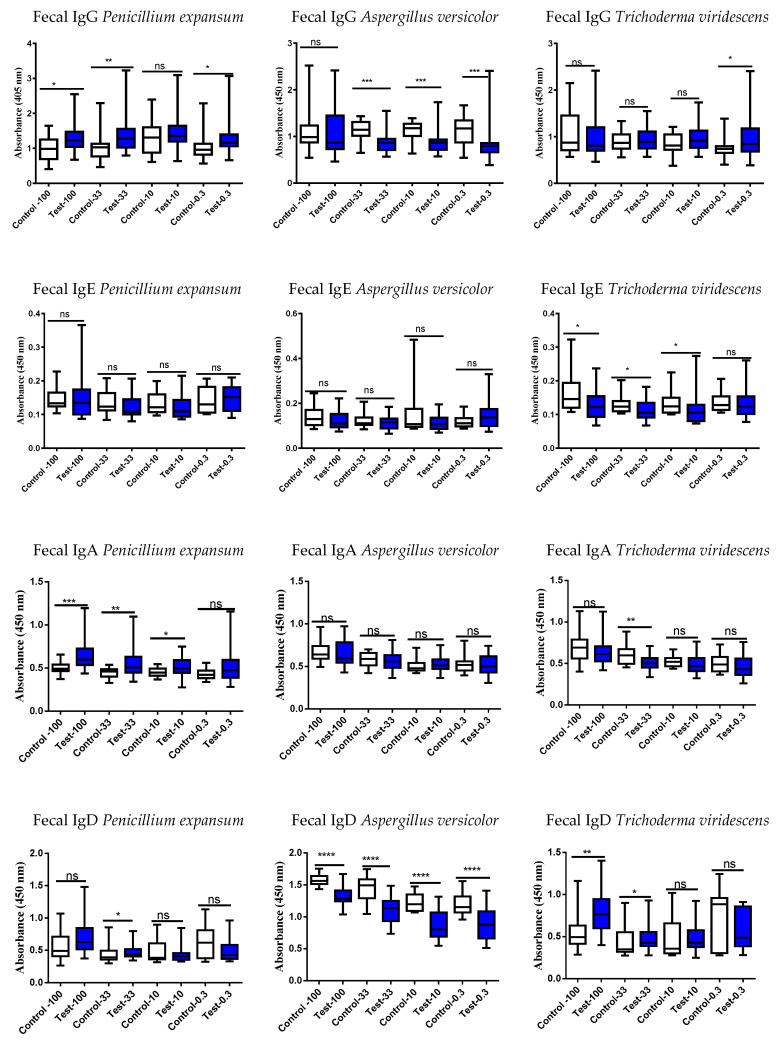
Fecal Ig responses against extracts of *Penicillium expansum, Aspergillus versicolor*, and *Trichoderma viridescens.* Test = exposed subjects from the target, toxic building; control = unexposed subjects. It is commonly known that IgA plays a marked role in intestinal immunology, but there were nonsignificant IgA levels against *A. versicolor* and *T. viridescens* strains. The IgG and IgD results of *A. versicolor* revealed markedly reduced responses in the exposed subjects. Note that the protein dilutions of the fecal samples are 100, 33, 10, and 0.3 µg/mL. Boxplots indicate median values with 95% confidence intervals; statistically significant * *p* < 0.05, ** *p* < 0.01, *** *p* < 0.001 and **** *p* < 0.0001 when the study persons were compared to controls. Ns = non-significant.

**Figure 4 jof-09-00332-f004:**
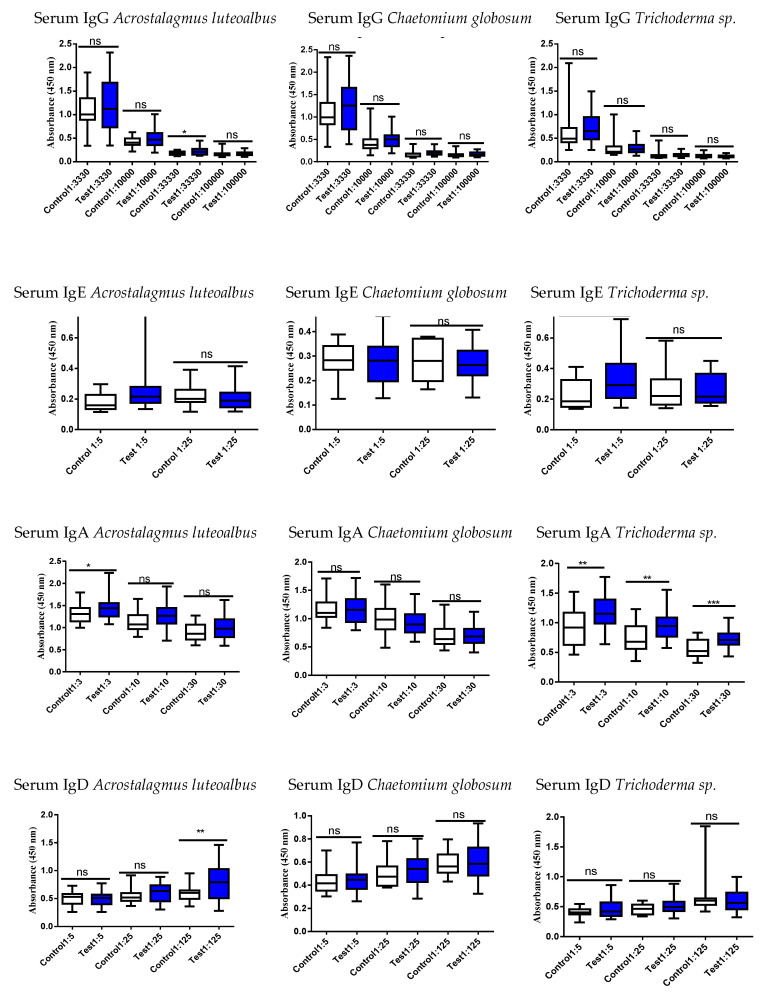
Serum Ig responses against *Acrostalagmus luteoalbus, Chaetomium globosum* and *Trichoderma* sp. *(=atroviride 14/AM).* Serum levels of IgE and IgA against *A. luteoalbus* and *Trichoderma* sp. were increased in exposed subjects, but the results were otherwise unable to differentiate between exposed and control subjects. Note that there is no difference between subjects in all *C. globosum* Ig results. Although this highly toxic isolate of *C. globosum MTAV35* used was not from these buildings, it was used as a reference strain. There was an abundance of *C. globosum* strains in the target building, but these were not studied [20,21]. Boxplots indicate median values with 95% confidence intervals; statistically significant * *p* < 0.05, ** *p* < 0.01, *** *p* < 0.001 and **** *p* < 0.0001 when the study persons were compared to controls. Ns = non-significant.

**Figure 5 jof-09-00332-f005:**
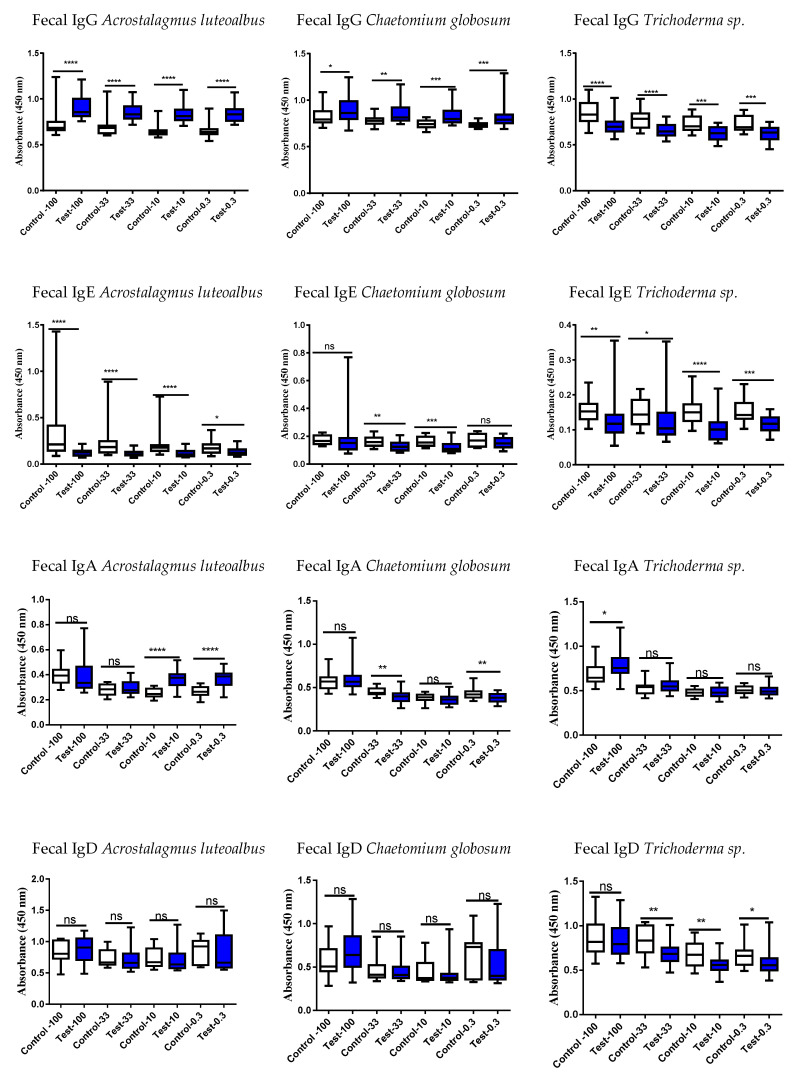
Fecal Ig responses against *A. luteoalbus, C. globosum MTAV35*, and *Trichoderma* sp. *(=atroviride 14/AM).* Fecal IgG levels of both *A. luteoalbus* and *C. globosum MTAV35* were increased in exposed subjects. With respect to *Trichoderma* sp. IgE values, the extensive variation complicates the interpretation but the variation in the results is not attributable to a single study subject. Boxplots indicate median values with 95% confidence intervals; statistically significant * *p* < 0.05, ** *p* < 0.01, *** *p* < 0.001 and **** *p* < 0.0001 when the study persons were compared to controls. Ns = non-significant.

**Figure 6 jof-09-00332-f006:**
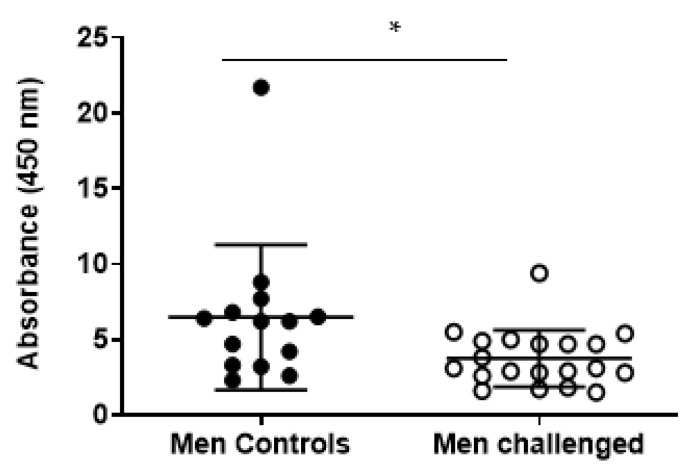
Statistical difference in the levels of the novel inflammation marker MRP8/14 (serum calprotectin) when exposed men are compared to control men, *** denotes significant difference, *p* = 0.0145.

**Figure 7 jof-09-00332-f007:**
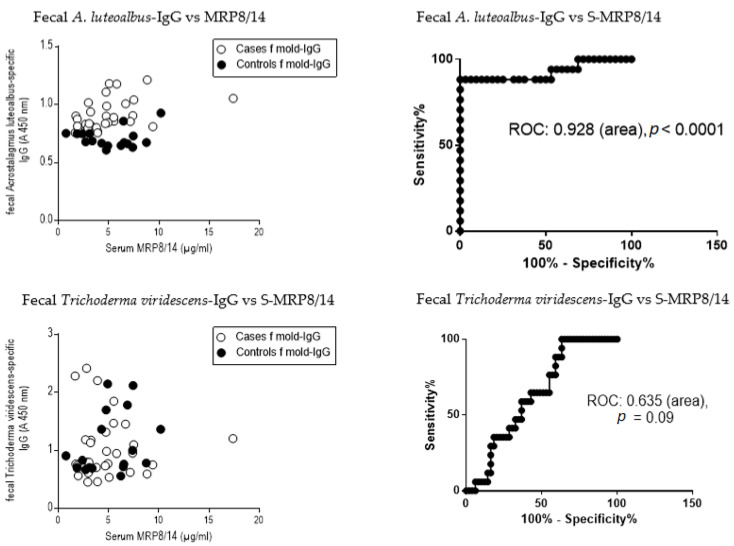
Receiver operating curves ROC results of fecal IgG vs. inflammatory marker MRP8/14 (calprotectin), when fecal coating is 100 µg/mL. The ROC curve analyses indicated the diagnostic performance of the specific Ig against the subject’s inflammation status, as determined by MRP8/14 levels. The best resolution results of microbial-specific Igs are evident in Figure 5 and the poorest resolution result in Figure 3. The best resolution result of Figure 5 from the first row on the left (fecal *A. luteoalbus*-specific IgG) was plotted against the concentration of MRP8/14, a marker of inflammation, in exposed subjects (open circles) and controls (black circles). On the top right-side is the ROC curve with the specificity and sensitivity calculated. The lower row shows as an example of how the respective results of *T. viridescens*-specific fecal IgG results do not distinguish mold-exposed subjects from controls (Figure 2 right top), and the ROC curve value is in line with the conclusion that fecal *T. viridescens*-specific IgG results were of no value when evaluating whether the subject had been exposed to harmful toxins.

**Figure 8 jof-09-00332-f008:**
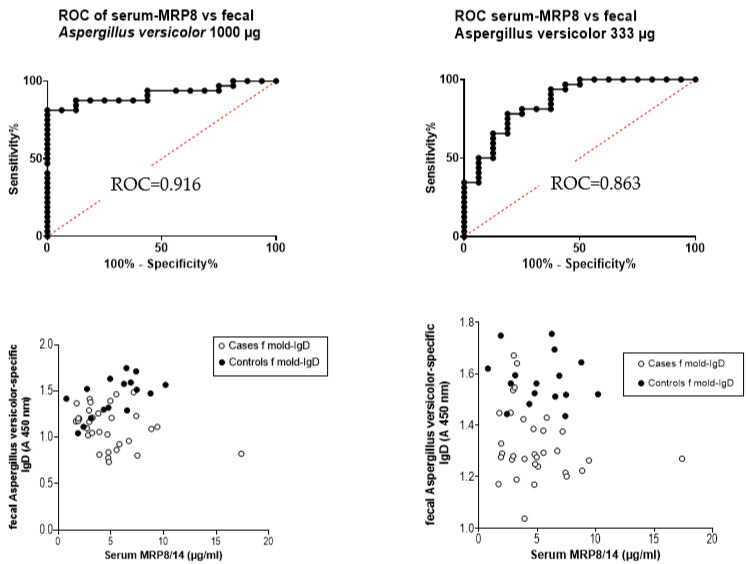
Impact of the protein levels in the coating of the microtiter wells for ROC assays. All the results are from IgD responses of fecal samples against *Aspergillus versicolor* when plotted against the serum inflammation marker MRP8/14. The amount of the coating extract was studied by conducting the assay either in 1000 or 333 µg/mL of *A. versicolor* extract.

**Table 1 jof-09-00332-t001:** Microbial strains isolated from the target building, expect for the *Chaetomium globosum MTVA35*, which was from Oulu University. This target building was selected based on the previous publications and results of the toxicity evaluations.

Code	Strain	Identification	Ref	Source	Toxicity in Cell Test Cytotoxicity *		
					Porcine Sperm Motility Inhibition Test	PK-15 **	MNA ***
POB8	*Acrostalagmus luteoalbus*	Andersson et al., 2021	[18]	First floor Insulation board from the wall	+	+	
MH33	*Aspergillus* Section *versicolor*	Salo et al., 2019	[16]	First floor settled room dust	−	+	
P61	*Penicillium expansum*	Salo et al., 2019	[15]	Insulation board from the wall	+	+	+
14/AM	*Trichoderma atroviride*	Castagnoli et al., 2018	[19]	From the mechanical ventilation outlet filter	+	+	+
Sip32	*Trichoderma viridescens*	Castagnoli et al., 2018	[19]	Vacuum cleaner from the dust bag	+	+	+
MTAV35=SZMC 26539	*Chaetomium globosum*	Salo et al., 2020 Kedves et al., 2021	[20,21]	Dust sample from Oulu University, Finland	+	+	+
MH5=SZMC24456	*Chaetomium globosum*	Salo et al., 2020 Kedves et al., 2021	[20,21]	Settled dust	+	+	+

Plus or minus mark indicates that the mold strain is toxic or not in the particular cell tests. * Cytotoxicity measured as inhibition of proliferation, cells were inert or were unable to proliferate. ** Porcine kidney epithelial cells (PK-15), *** murine neuroblastoma cells (MNA). *Chaetomium globosum MTAV35* is an indicator strain, which was not isolated from the target nor from the control building but was used as a control; note that the target building did contain several strains of *C. globosum MH5* and *MH52* [20,21], which were not studied here. Ref = references for these results are from earlier publications.

**Table 2 jof-09-00332-t002:** Self-reported symptoms reported from employees in the target (29/32 subjects, 91%) and control buildings (4/18 subjects, 22%).

Sex	ID	Exposed Cases
M	H27	Airway problems.
M	H28	Asthma diagnosis.
F	H29	Gluten-intolerance found years ago. Severe pains.
F	H30	No symptoms reported.
M	H31	Chronic rhinitis.
F	H32	Dyspnea, fatigue, blood pressure and asthma medication, severe eye symptoms, arthritis.
F	H33	Severe skin inflammation like a smallpox over the body, hot skin abscesses, drug-resistant
		except for a response to cortisones.
F	H34	Nasal congestion, skin symptoms.
M	H35	Fatigue, nasal congestion.
F	H36	Rhinitis, cold several times a year.
F	H37	Sinusitis, gastric symptoms, ear and sinus infections; chronic rhinitis that only disappeared
		during the summer vacation.
M	H38	Insomnia, body temperature normally 34 °C, mycoplasma, muscle weakness, neck pain,
		short-term memory problems, thyroid symptoms.
M	H39	Dyspnea, hoarseness, insomnia. Cholesterol medication.
M	H40	No symptoms.
M	H41	Hoarseness, fatigue, stinging eyes.
M	H42	Before medication had rhinitis, sore throat, cold twice a month. In wintertime, there were
		pimples on the back. Currently being treated with an antihistamine, cortisone, and beta-
		agonists. Can only spend max 1–2 h in his workroom.
M	H43	No symptoms.
M	H44	Hoarseness and often cough; occasional muscle twitching.
F	H45	Itchy hands, dry throat, nosebleeds, short duration memory problems, fatigue, redness and
		itching of the throat area. Cannot stay in the building longer than one day at a time due to
		the itchiness.
M	H46	Many nosebleeds, arthritis in every joint, worsening when walking at a rapid pace, urticaria,
		angioedema, vasculitis.
M	H47	Fatigue and difficulty in falling asleep. Allergy symptoms when cleaning this building.
		Irritation of skin from concrete dust and oils. Symptoms from dust: rhinitis, congestion,
		joint pain.
F	H48	Crohn’s disease, sclerosing cholangitis. Dryness of the nose and constant sneezing.
M	H49	Rheumatoid arthritis, diverticulitis.
F	H50	Persistent cough, dyspnea, low oxygen saturation, chronic polyposis; being treated with
		asthma medication.
M	H51	Lump in the larynx, hoarseness, and cough. The symptoms disappeared after a 4-week
		vacation, and returned when the individual came back to work in the target building.
M	H52	Skin rash, glucocorticoids in use.
M	H53	Diagnosed with intestinal problems.
F	H54	Possibly memory problems.
M	H55	Alternating dyspnea, rhinitis in the mornings, severe hair loss.
M	H56	Diagnosed asthma. Uses inhaled glucocorticoids and beta-agonists, frequent colds.
M	H57	High blood pressure, dysrhythmias. Repeated symptoms suggestive of asthma, daily dry
		throat, hoarseness, cough. Joint pains that pass rather quickly.
F	H58	Reflux, hoarseness, irritated throat, associated with reflux.
**Sex**	**ID**	**Controls**
M	H59	No symptoms nowadays. Exposed to molds 19 years ago.
F	H60	No symptoms.
F	H61	Migraine, sensitive stomach.
F	H62	No symptoms from the control building. Mild food intolerances.
M	H63	No symptoms.
M	H65	Spring cold, no symptoms.
M	H66	No symptoms.
F	H67	No symptoms now, but when employed in another building for 2–3 years, she experienced
		nosebleeds and migraines, intestinal problems; symptoms relieved when
		she stopped working in that building.
M	H68	Celiac disease, no other symptoms.
M	H69	Gastric problems.
M	H70	No symptoms.
M	H71	No symptoms.
M	H72	No symptoms.
M	H73	No symptoms.
M	H74	No symptoms.
F	H75	No symptoms.
F	H76	No symptoms.
M	H77	No symptoms.

**Table 3 jof-09-00332-t003:** Indoor air water samples from the target building. Samples were collected by two different building inspectors, 1 and 2. The difference between the results of Room 266.1 can be explained due to the different positions of the collector devices. Inspector 2 took samples close to the incoming air vent, whereas those from inspector 1 were close to the outgoing air vent. Results are provided as mean ± stdev; *p* < 0.05 means that the result is statistically significant; *p* > 0.05 means that the result is not statistically significant.

Sampling Site	Inspector	Toxicity in THP-1 Macrophage (%)	Classification of
	(Sample Collector)	at 10% Sample Concentration	Toxicity: Yes/No
		Mean ± Stdev, *p*	
Room 264.1. A	1	6.20 ± 2.90 *p =* 0.009	Yes
Student classroom	2	7.10 ± 3.30 *p =* 0.004	Yes
Room 265.1.	1	4.70 ± 2.40 *p =* 0.034	Yes
Student classroom with students	2	7.30 ± 2.90 *p =* 0.003	Yes
sitting in the classroom			
Room 266.1.	1	7.80 ± 2.90 *p =* 0.002	Yes
Empty classroom, outgoing air			
Room 266.1.	2	2.00 ± 4.00 *p =* 0.375	No
Empty classroom, ingoing air			
Outdoor air at door EA.1	1	0.80 ± 3.90 *p =* 0.709	No
Outdoor air at PS.1	2	2.50 ± 2.80 *p =* 0.240	No

**Table 4 jof-09-00332-t004:** The indoor air water samples from the control building did not induce toxicity in THP-1 tests, and sampling was performed by one inspector. Results are provided as mean ± stdev; *p* < 0.05 means that the result is statistically significant; *p* > 0.05 means that the result is not statistically significant.

Sampling Site	Toxicity in THP-1 Macrophages (%) at 10% Sample Concentration Mean ± Stdev, *p*	Classification of Toxicity Yes/No
TU3 Classroom front area	−0.50 ± 7.90 *p =* 0.872	No
TU5 Classroom middle area	3.50 ± 5.70 *p =* 0.246	No
TU7 Classroom front area	0.80 ± 4.60 *p =* 0.790	No
A.1. Outdoor area	2.30 ± 4.30 *p =* 0.420	No

**Table 5 jof-09-00332-t005:** Comparison of the results; exposed vs. controls in data extracted from Figure 2 vs. Figure 3 and Figure 4 vs. Figure 5. The red color indicates an increase in the Ig response of exposed subjects when compared to the controls. The yellow means that there is no difference between the study groups, whereas green refers to a decrease in the level of that Ig when the exposed subjects are compared to controls. In general, it appears that if there is an increase in the Ig response in serum, the Ig in question does not display an increase in the fecal Ig response or vice versa. An exception (x) is found in the IgA level of *A. luteoalbus*, in which only the highest serum and lowest fecal IgA levels of the studied sample dilutions show increased results in exposed vs. controls. Moreover, in *A. versicolor*, the IgD results are decreased in the exposed both in serum and fecal results.

	*P. expansum*		*A. versicolor*		*T. viridescens*		*A. luteoalbus*		*C. globosum*		*T. atroviride* sp.	
	Ser. Ig	Fecal Ig	Ser. Ig	Fecal Ig	Ser. Ig	Fecal Ig	Ser. Ig	Fecal Ig	Ser. Ig	Fecal Ig	Ser. Ig	Fecal Ig
IgG		x	x					x		x		
	x				x	x	x		x		x	
				x								x
IgE	x		x		x		x				x	
		x		x					x			
						x		x		x		x
IgA		x					x	x			x	
				x		x			x			x
	x		x		x					x		
IgD						x						
	x	x			x		x	x	x	x	x	
			x	x								x

## Data Availability

Not applicable.

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
