# Peer review of "In Search of Clinical Markers: Indicators of Exposure in Dampness and Mold Hypersensitivity Syndrome (DMHS)"

_jof, 2023, doi:10.3390/jof9030332_

Round 1

Reviewer 1 Report

-ROC should be completely written in the first appear.

-Key words are to much (mold, diagnosis) can be deleted.

-Ethical concern about human samples....

-page 3 first line.......as documented by different investigators 14-16.

-Please do not use We or our along over the test.

-Fig 2  not bold.

-conclusion not reflect the whole- study, Ig assays should be added to conclusion.

Author Response

We thank the Reviewer 1 for valuable critics of the manuscript. 

-ROC should be completely written in the first appear. This has now been added to the abstract of the manuscript.

-Key words are to much (mold, diagnosis) can be deleted. Those two are now deleted.

-Ethical concern about human samples.... We do not find this place, can you please help us with this?

-page 3 first line.......as documented by different investigators 14-16. This has been changed now.

-Please do not use We or our along over the test. We have done our best for this. Hopefully you accept if some place is not changed. For instance line 244, 249, 442 & 446 since we are the only ones in the world who has done that. The word "our" is on lines 249 and 251 for the same reason.

-Fig 2  not bold. This part we do not understand. All the figure legends start with bold text, but then the text is not in bold. As for the bold or not bold text parts in the figure titles, I find it difficult to make any changes, since due to that it has now taken years to get this article processed, I have mean while lost the program GraphPad Prism from my computer. 

-conclusion not reflect the whole- study, Ig assays should be added to conclusion. More results from the Ig's have now been added to the conclusion.

Reviewer 2 Report

The reviewer has commented as following for the authors.

Line 39: The font of “subjects” can be checked.

Line 41: Larger font size of the reference numbers are needed. Please check the complete manuscript.

Line 64: names of the species should be listed in alphabetical order. Please check the complete manuscript.

Table 1: Three-line table is suggested. Please check the other tables.

Line 101: Font of Chaetomium globosum

Line 113: A link for the web site.

Line 130: International System of Units should be used in manuscript preparation. It can be “μL”, not “μl”, same as “mL” and “mM” (line 138). Please check the complete manuscript.

Line 126: A “24h” in this line. A “3 hours” in line 130.

Table 2: This table seemed confusing. A header between H58 and H59. A cross on H64.

Line 245: It should be 3%–5%. Please check the complete manuscript.

Table 3: P for standard values should be in italic. Please check this point.

Pages 10 and 12: Why large area of these pages are in blank.

Line 284 to 285: Different fonts.

Lines 102 and 286: The reviewer suggests the authors not to use C. globosum in the bodytext, anyway, there is Cladosporium herbarum. A full spelling for C. globosum is better for avoiding the possible confusion.

Author Response

We thank the Reviewer 2 for helpful suggestions to this manuscript.

Line 39: The font of “subjects” can be checked. It has been changed now.

Line 41: Larger font size of the reference numbers are needed. Please check the complete manuscript. This has been changed, thank you.

Line 64: names of the species should be listed in alphabetical order. Please check the complete manuscript. This is now in alphabetical order.

Table 1: Three-line table is suggested. Please check the other tables. Based on my earlier publication in the Journal of Fungi, I think it is allowed to have more than just three-line table. Otherwise it would rather difficult to produce all the needed information.

Line 101: Font of Chaetomium globosum. This was changed.

Line 113: A link for the web site. This is now in the EndNote and list of references.

Line 130: International System of Units should be used in manuscript preparation. It can be “μL”, not “μl”, same as “mL” and “mM” (line 138). Please check the complete manuscript. These were now changed. 

Line 126: A “24h” in this line. A “3 hours” in line 130. These have now been changed.

Table 2: This table seemed confusing. A header between H58 and H59. A cross on H64. The H64 is deleted now.

Line 245: It should be 3%–5%. Please check the complete manuscript. The space before % has now been deleted.

Table 3: P for standard values should be in italic. Please check this point. These were now changed.

Pages 10 and 12: Why large area of these pages are in blank. This was an accidental page break and has now been deleted. The figures from 2-5 need full page space, so there was a technical need to use forced page break.

Line 284 to 285: Different fonts. These were now changed.

Lines 102 and 286: The reviewer suggests the authors not to use C. globosum in the bodytext, anyway, there is Cladosporium herbarum. A full spelling for C. globosum is better for avoiding the possible confusion. This is now changed to the abstract, but I think it should be enough since C. globosum is then presented like all the other mold strains, and it does differentiate from C. herbarum.